# An Omics View of Emery–Dreifuss Muscular Dystrophy

**DOI:** 10.3390/jpm10020050

**Published:** 2020-06-15

**Authors:** Nicolas Vignier, Antoine Muchir

**Affiliations:** INSERM, Center of Research in Myology, Institute of Myology, Sorbonne University, 75013 Paris, France; n.vignier@institut-myologie.org

**Keywords:** *LMNA*, Emery–Dreifuss muscular dystrophy, Omics

## Abstract

Recent progress in Omics technologies has started to empower personalized healthcare development at a thorough biomolecular level. Omics have subsidized medical breakthroughs that have started to enter clinical proceedings. The use of this scientific know-how has surfaced as a way to provide a more far-reaching view of the biological mechanisms behind diseases. This review will focus on the discoveries made using Omics and the utility of these approaches for Emery–Dreifuss muscular dystrophy.

## 1. Introduction

To understand the complexity of systems biology, Omics’ technologies adopt a holistic view. In this, in opposition to hypothesis-generating experiments, no rationale is known, but instead, biological inputs are acquired and analyzed to delineate a hypothesis that can be then tested. Omics technology can be used not only to decipher physiological conditions but also in disease states, where they have a key role in diagnosis, as well as promoting our knowledge of the development of diseases [1]. Omics approaches to conditions, such as muscular dystrophies, are being used for small molecule therapy discovery by isolating innovative targets for drug development [2]. The scope of this review is to provide an overview of the Omics approaches and their application in Emery–Dreifuss muscular dystrophy research.

## 2. Emery–Dreifuss Muscular Dystrophy

Muscular dystrophies are characterized by the progressive weakness and degeneration of the skeletal muscle system, which may or may not be associated with cardiac impairment, leading to loss of mobility, and swallowing and respiratory difficulties. Death originates from respiratory defects or heart failure. Muscular dystrophies are a heterogeneous group of inherited disorders, and they differ in the distribution of affected muscles, the rate of muscle weakness progression and the age of onset [3]. The development of molecular genetic mapping techniques has shown that these disorders are genetically heterogeneous, and more than 50 genes have been identified as causing muscular dystrophies [4].

In the 1960s, an X-linked muscular dystrophy associated with contractures, which was first diagnosed as a benign variant of Duchenne muscular dystrophy, was reported [5,6]. In the 1980s, Alan Emery re-investigated the original family and reported that cardiomyopathy was a significant feature of the disease, which was thereafter called Emery–Dreifuss muscular dystrophy (EDMD) [7]. In EDMD, the onset of symptoms occurs within the first decade of life [7]. Contractures of the elbows, neck extensor muscles and Achilles’ tendons appear to be the first symptoms of the disease, and occur before muscle weakness and wasting. The progressive muscle degeneration begins during the end of the second decade of life, in a humeroperoneal distribution. Cardiac alteration begins during the teenage years, with no link to the severity of the muscular dystrophy [7,8,9,10]. Over time, dilated cardiomyopathy develops, and is associated with severe ventricular tachydysrrhythmias. Sudden cardiac death is frequent, and an implantable defibrillator can be lifesaving [10,11].

## 3. Application of Omics Approaches for Emery-Dreifuss Muscular Dystrophy

### 3.1. Omics and Diagnosis

In the 1990s, a positional cloning study—a technique for the positioning of a trait-associated gene within the genome—showed that mutations in *EMD*, and *LMNA* cause the X-linked [12] and autosomal dominant [13] forms of EDMD, respectively [14]. EMD encodes emerin, which is a transmembrane protein of the nuclear envelope. *LMNA* encodes nuclear lamins A and C, which are intermediate filament proteins associated with the nuclear envelope. Until recently, genetic screening for EDMD was performed with Sanger sequencing of the exons and intron-exon regions of both the *EMD* and *LMNA* genes. Since then, diagnosis of EDMD has been made available if the clinical signs are suggestive or if a family member is known to have EDMD. However, classical DNA sequencing methods have many shortcomings (time and cost) hampering their use for diagnosis. By contrast, as introduced in 2005, next-generation sequencing is a potent novel technology and a cost-effective method that has completely revolutionized the field [15,16]. Full exome sequencing is routinely used in clinical diagnostic laboratories to identify pathogenic variants in a given patient at a reasonable cost [17,18,19,20]. A new next-generation sequencing approach to identify potential new candidate EDMD genes has also recently been tested [21].

Subsequently, *LMNA* mutations have been shown to cause other striated muscle diseases, i.e., dilated cardiomyopathy [22], limb-girdle muscular dystrophy type 1B [23] and congenital muscular dystrophy [24]. Limb-girdle muscular dystrophy type 1B and dilated cardiomyopathy can occur in the same families as subjects with EDMD, and can be therefore be considered variants of the same clinical entity. This supports the concept that modifier genes arouse the severity and the peculiar symptoms. To map the modifier locus, microsatellite markers were genotyped in a large French family, where patients carrying the same *LMNA* mutation exhibited phenotypic variability [25]. The linked DNA region harbors two candidate modifier genes, *DES* and *MYL1*, encoding desmin and light chain of myosin, respectively, thus providing insights for the natural history and the physiopathology of EDMD.

### 3.2. Omics and Abnormal Cellular Signaling

The mechanisms by which mutations in *EMD* and *LMNA* cause muscular dystrophies are poorly understood. A few models have been proposed to explain the physiopathology of EDMD [26]. The model called the ‘mechanical stress hypothesis’ relies on the premise that striated muscle is steadily subjected to mechanical strains. Abnormalities in nuclear envelope composition may imply a weakening of the nucleus, which could represent an initial step in the chain of events leading to EDMD.

The ambition of transcriptomics studies is to pinpoint genome-wide changes and to expose coordinately organized gene networks [27,28]. Gene expression profiles associated with EDMD have been studied in several experimental models using various technologies. Mouse models have been helpful in deciphering mechanisms involved in the pathogenesis of EDMD, as well as for opening pharmacological therapies perspectives. The development of *Lmna*^-/-^ mice by Sullivan and colleagues was the first animal model of the disease [29]. The mice expressing a truncated peptide, lamin A delta8-11 [30], develop cardiomyopathy and skeletal muscle wasting reminiscent of human EDMD. Then, other mouse models of EDMD were generated. These are knock-in mice that express A-type lamins with the p.H222P [31], p.deltaK32 [32] and p.N195K [33] residue substitutions. Arimura et al. developed *Lmna* knock-in mice carrying the p.H222P mutation that was identified in the human *LMNA* gene in an EDMD family [31]. This mutation was also chosen because it putatively dramatically altered the coil-coiled organization of A-type lamins, based on in silico analysis. This was the first *Lmna* mouse model mimicking human EDMD from the gene mutation to the clinical symptoms. Two separate groups have generated *Emd* null mice [34,35]. These mice have subtle motor coordination abnormalities, with a prolongation of atrioventricular conduction time in *Emd* null mice [35]. Notwithstanding, these animal models may not reflect the “natural” human condition in terms of physiological mechanism and genetic outlook. Induced pluripotent stem cell (iPSC) technology represents a means to surmount these shortcomings, allowing the generation of any cell type through peculiar differentiation protocols [36]. Tissue-specific in vitro models of EDMD have been created from iPSCs that recapitulate traits of the disease [37,38,39,40,41,42].

To explore the pathogenesis of cardiomyopathy associated with EDMD, we carried out a genome-wide RNA expression analysis of hearts from *Lmna*^p.H222P/H222P^ mice and *Emd* knockout mice, two mouse models of EDMD. This analysis revealed changes in the expression of genes encoding proteins in mitogen-activated protein (MAP) kinases, Wnt/β-catenin, AKT/mTOR and transforming growth factor (Tgf)-β signaling pathways in the mutated models [43,44,45,46,47,48]. Using RNA-sequencing technology on *Lmna*^-/-^ mice, Auguste and colleagues showed that the FOXO signaling pathway impacted different signaling pathways, i.e., NFκB, TNFα, P53 and OxPHOS signaling pathways and biological processes, i.e., apoptosis, sustaining the cardiac phenotype associated with EDMD [49]. Furthermore, whole genome expression analysis of the primary cells of EDMD patients showed aberrant activity of unfolded protein response signaling [50]. In a cardiac specific expression of *Lmna* p.D300N, the Marian group showed using bulk RNA sequencing strategy that an increase of DNA damage response/TP53 pathway was contributing to the pathogenesis of cardiomyopathy associated with EDMD [51]. All these datasets led to the hypothesis of a model of how abnormalities of A-type lamins and emerin may lead to EDMD [52]. Using a transcriptomic approach from regenerating skeletal muscle from emerin-deficient mice, Melcon et al. [34] have shown delayed myogenic differentiation, which is regulated by *Rb* and *MyoD* genes. Since then, small molecules have been used by others to rescue the impaired myogenic differentiation in emerin deficiency, which could represent a potential strategy for improving the muscle wasting phenotype seen in EDMD [53]. Hence, abnormalities in satellite cell behavior may be responsible in part for the skeletal muscle disease in EDMD.

The mechanisms that bridge the *LMNA* genetic defects to malignant arrhythmias [54] are unknown. To better understand this phenotype in EDMD, Dr. Wu’s group modeled the disease in vitro using patient-specific iPSC-derived cardiomyocytes (iPSC-CMs) carrying *LMNA* frameshift p.K117fs mutation [42]. They showed an abnormal activation of the platelet-derived growth factor (PDGF) signaling in *LMNA* p.K117fs iPSC-CMs. The inhibition of the PDGF signaling improves the arrhythmic phenotype of mutated iPSC-CMs, opening novel therapeutic perspectives for the treatment of EDMD.

All these studies contributed to a better knowledge of the functional and molecular mechanisms of the disease. We can expect that new findings will design applications of iPSC-models to pharmacological testing in striated muscle-specific contexts [55], making the technology available to patients.

### 3.3. Omics and Chromatin Regulation

Among the models proposed to explain how EDMD phenotypes arise, the ‘gene-expression model’, posits an effect of mutated lamin A/C on the transcription activity of genes and/or pathways that could impact striated muscle-homeostasis process. According to this model, it has been described that A-type lamins interact with heterochromatin regions called Lamin-Associated Domains (LADs). These LADs play a role for chromatin organization and gene expression regulation [56,57,58,59]. It has been shown that the LADs are re-organized in EDMD steering modifications of the epigenetic program, ultimately driving the loss of myogenic differentiation [60]. This has been recently shown in an elegant manner by Bianchi and colleagues on a murine model of EDMD [61]. The authors described an abnormal positioning of polycomb proteins, which are epigenetic repressors involved in cell identity. This causes impairment in self-renewal, deficiency of cell identity and the early exhaustion of the quiescent satellite cell pool, demonstrating that muscular dystrophy in EDMD can be partially caused by epigenetic dysfunctions of muscle stem cells [61]. Mewborn and colleagues showed that the *LMNA* p.E161K mutation perturbed the positioning and compaction of chromosomal domains in primary fibroblasts, resulting in an altered gene expression profile [62]. Another study focused on the organization of LADs in EDMD using a multi-Omics approach (Chip-Seq/RNA-sequencing) in explanted hearts from five patients carrying *LMNA* mutations [62]. LADs were redistributed, ensuing in a functional chromatin state in mutated hearts, suggesting the loss of specific functional chromatin binding. This aberrant distribution impacted both the gene expression profile and CpG methylation. An integrated analysis showed the combined role of LADs and CpG methylation in the regulation of gene expression, and identified numerous transcription factors involved in biological processes such as cell death/survival, cell cycle and metabolism [63]. Bertero and colleagues showed at the same time, using genome-wide chromosome conformation capture (HiC) analyses on iPSC-CMs carrying *LMNA* p.R225X mutation [64], a slight chromatin compartment dysregulation (around 1% of the genome). RNA-sequencing of these altered chromatin domains revealed the abnormal up-regulation of only a handful of genes, among which was *CACNA1A*. This latter encodes a subunit of a calcium channel, whose abnormal expression might partially explain the electrophysiological and contractile aberrations observed in the mutant hiPSC-CMs. The authors showed that pharmacological treatments to prevent both the electrophysiological and contractile alterations helped improve the abnormal phenotypes described in the mutated iPSC-CMs. This suggests that *CACNA1A* may be a good therapeutic target to reverse the cardiac abnormalities in EDMD patients. However, the work by Bertero et al. showed only minor alterations in chromatin compartmentalization in mutated hiPSC-CMs, a finding that challenges the aberrant gene-expression model [64].

### 3.4. Omics and Metabolism

Contracting incessantly, the heart demands a lot of energy to ensure optimal contractile function. Research has demonstrated that the high requirements of the heart are satisfied by a preference for the oxidation of fatty acids. Studies have demonstrated that the failing heart deviates from its inherent profile and relies heavily on glucose metabolism, primarily achieved by acceleration in glycolysis. To gain further insight into the molecular mechanism ruling this disease, scientists have studied the cardiac metabolic rates in *Lmna*^-/-^ mice. West and colleagues described a targeted metabolomics assay that quantifies metabolites relevant to cardiac metabolism [65]. The assay demonstrates that the *Lmna*^-/-^ mouse heart has decreased metabolites associated with the citric acid cycle and fatty acid oxidation [65]. This corroborated another study, which showed that activated AKT/mTOR signaling reduces tolerance to energy deficits in hearts from *Lmna*^p.H222P/H222P^ mice [45]. A rapamycin analog that blocks AKT/mTOR activity has been used to prevent the progression of cardiomyopathy in *Lmna*^p.H222P/H222P^ mice [45]. These works highlighted that the heart is unable to compensate for increased or fluctuating energy demand and, over time, develops dilated cardiomyopathy in EDMD. This metabolic remodeling probably represents an adaptive cardio-protective mechanism that can help improve contractile function, thus slowing the progression of EDMD and improving prognosis. As such, metabolic modulators, which have the potential to shift myocardial substrate utilization from fatty acids toward glucose metabolism, may have a place in the management of patients. Some of these modulators have already been investigated as treatment for cardiomyopathy, with some beneficial effects [66,67]. It would be relevant to test their efficacy in EDMD.

Moreover, West and colleagues showed increased responses to oxidative stress and reactive oxygen species (ROS) exposure in *Lmna*^-/-^ mouse hearts [65]. ROS are small, short-lived signaling molecules that mediate various cellular responses. Based on this, we recently showed that N-acetyl cysteine treatment reduces cardiac oxidative stress injury and ameliorates contractile dysfunction in *Lmna*^p.H222P/H222P^ mice [68].

### 3.5. Omics and Biomarkers

Omics are powerful tools to identify diseases’ molecular biomarkers. Molecular biomarkers are molecules with particular biophysical properties, the quantities of which are measured in biological samples, and are of critical importance in the support of the clinical diagnosis of pathology, the monitoring of its time course and the evaluation of the impact of therapeutic approaches. Molecular biomarkers have to be quantified from patients in the least invasive manner possible. Circulating molecular biomarkers in body fluids, i.e., blood, plasma, serum or urine, are thus the main interest. Proteomic, transcriptomic and metabolomic analysis have been driven in many diseases to identify such biomarkers [69,70].

To identify circulating microRNA as molecular biomarkers for EDMD, the microRNA transcriptome (miRnome) from plasma of *Lmna*^p.H222P/H222P^ mice was screened [71]. A specific and distinctive microRNA expression profile was identified, and three of the dystromirs (mir-133b, mir-133a and mir-1) were downregulated in these mice. Furthermore, two microRNAs were upregulated (mir-146b and mir-200a) and six microRNAs were downregulated (mir-130a, mir-133a, mir-133b, mir-1, mir-151-3p, mir-339-3p) in *Lmna*^p.H222P/H222P^ mice compared with wild type animals [71].

## 4. Conclusions

Omics technologies have hastened the identification of genetic mutations associated with EDMD, and have unveiled the existence of rare variants and modifier genes that might establish the phenotypical heterogeneity of the disease. Transcriptomic studies have uncovered alterations in signaling mechanisms causing some of the symptoms in EDMD, but are restricted to experimental models (in vitro and in vivo) with technical and theoretical shortcomings. Furthermore, although the number of parameters being measured has increased with Omics technologies, the number of biological and methodological replicates has not. In addition, because of the large number of measurements and the limited number of subjects, unique problems arise in Omics studies involving statistics and bias. A single Omics technique will only capture changes in a subset of biological cascades; it cannot provide a systemic understanding of the complexity of systems biology. The integration of multiple Omics data sets promises a substantial improvement, through an increase of information and, especially, systemic understanding. Therefore, much work is needed before using this research in the clinic. All of this will depend on integrative collaborations among physicians and scientists that will be essential for major breakthroughs for both the diagnosis and treatment of EDMD.

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
