# Peer review of "An Omics View of Emery–Dreifuss Muscular Dystrophy"

_jpm, 2020, doi:10.3390/jpm10020050_

Round 1
Reviewer 1 Report
Authors have very systematically summarized the role of multiple high throughput plate forms such as transcriptomics, genomics, etc. in a better understanding of Emery-Dreifuss muscular dystrophy. It will be really great if authors ca add additional comments on the shortcomings of some of the studies and how other comics technologies can complement or support the findings, will be great. Authors should also include some metabolomics studies about this disease or its potential application in understanding of Emery-Dreifuss muscular dystrophy.
Author Response
Please see the attachement

Reviewer 2 Report
The authors describe a nice review about EDMD and "OMICS2 with the potentiality of this technique.
I have only minor concerns:
1- the references should be updated:
Scott A . Haller . Muscle and Nerve ,2020;Bossone et al , Muscle and Nerve ,2020; Bertrand et al,cell,2020, Branch,JCI,20020
Author Response
Please see the attachment

This manuscript is a resubmission of an earlier submission. The following is a list of the peer review reports and author responses from that submission.